# Generalized decomposition of multivariate information

Thomas F. Varley [1,2] *

1 Department of Computer Science, University of Vermont, Burlington, VT, United States of America,
2 Vermont Complex Systems Center, University of Vermont, Burlington, VT, United States of America

* tfvarley@uvm.edu

## Abstract

Since its introduction, the partial information decomposition (PID) has emerged as a powerful, information-theoretic technique useful for studying the structure of (potentially higher-order) interactions in complex systems. Despite its utility, the applicability of the PID is restricted by the need to assign elements as either "sources" or "targets", as well as the specific structure of the mutual information itself. Here, I introduce a generalized information decomposition that relaxes the source/target distinction while still satisfying the basic intuitions about information. This approach is based on the decomposition of the Kullback-Leibler divergence, and consequently allows for the analysis of any information gained when updating from an arbitrary prior to an arbitrary posterior. As a result, any information-theoretic measure that can be written as a linear combination of Kullback-Leibler divergences admits a decomposition in the style of Williams and Beer, including the total correlation, the negentropy, and the mutual information as special cases. This paper explores how the generalized information decomposition can reveal novel insights into existing measures, as well as the nature of higher-order synergies. We show that synergistic information is intimately related to the well-known Tononi-Sporns-Edelman (TSE) complexity, and that synergistic information requires a similar integration/segregation balance as a high TSE complexity. Finally, I end with a discussion of how this approach fits into other attempts to generalize the PID and the possibilities for empirical applications.

## 1 Introduction

Since it was introduced by Claude Shannon in the mid-20[th] century, information theory has emerged as a kind of *lingua franca* for the formal study of complex systems [1]. A significant benefit of information theory is that it is particularly effective for interrogating the structure of interactions between "wholes" and "parts". This is a fundamental topic in modern complexity theory, as a defining feature of complex systems is the emergence of higher-order coordination between large numbers of simpler elements. Appearing in fields as diverse as economics (where economies emerge from the coordinated interactions between firms) to neuroscience (where consciousness is thought to emerge from the coordinated interaction between neurons), the question of higher-order structures in multivariate systems is of central importance in almost every branch of the so-called "special sciences" above physics. Information theory

**Competing interests:** The authors have declared
that no competing interests exist.

has been used to great effect in formalizing rigorous, domain-agnostic, definitions of "emergence" [2, 3] and exploring what the novel or unexpected consequences of emergence might be [4, 5]. These lines of research are active and fruitful, however, many of the techniques that have been used are limited to particular special cases, or specific kinds of dependency, which makes a general theory of higher-order information in complex systems difficult to achieve.

One of the most powerful tools in understanding the informational relationships between wholes and parts has been the partial information decomposition [6, 7] (PID), which decomposes the mutual information that a set of inputs collectively disclose about a target into redundant, unique, and synergistic "atomic" components (or higher-order combinations thereof). Since its proposal in 2011 by Williams and Beer, the PID has been fruitfully applied to a diverse set of complex systems, including hydrology [8], neuroscience [9, 10], medical imaging [11], the physics of phase transitions [12], machine learning [13, 14], economics [15], and clinical medicine [16, 17]. The PID has a handful of limitations, however. For instance, it requires designating a subset of elements as "inputs" and a single element as a "target." This can be a natural distinction in some cases (such as multiple pre-synaptic neurons that synapse onto a single downstream neuron [9]), however, this restriction makes a general analysis of "wholes" and "parts" more difficult, as the PID is inherently focused on how two different subsets of a system (the inputs and targets) interact. It would be useful to be able to relax the requirement of a firm input/target distinction and analyse the entire system *qua* itself.

The second limitation is that the mutual information refers to a very particular kind of dependency: it is an explicitly bivariate special case of the more general Kullback-Leibler divergence [18] and so may not be applicable to all circumstances. The mutual information is generally introduced as the information gained when updating to the true, joint distribution of elements from a hypothetical maximum-entropy prior distribution where all elements are independent (for more formal discussion, see below). While this is a natural comparison in many contexts, it is not the only useful definition of information. For example, it may not always make sense to have a prior of maximum entropy; perhaps ones initial beliefs about a system are more nuanced or informed by prior knowledge.

These limitations has been previously recognized: one attempt to relax the strong input/target distinction was the development of the partial entropy decomposition (PED) by Ince, and later Finn and Lizier (albeit under a different name) [19, 20]. Unlike the PID, which decomposes the joint mutual information, the PED uses the same logic to decompose the joint entropy of the whole system, without needing to classify subsets of the system. The PED has been used to analyse neural systems [21, 22], however it does not solve all the problems detailed above. While it does relax the input/target distinction, it does little to address the second limitation of PID: since it is a decomposition of entropy, not of information directly, it cannot be used as a general approach of multivariate information. The interpretation of the decomposition is completely different, and consequently, so is the behaviour. For example, for a set of two elements $X$ and $Y$, if $X \perp Y$ (i.e. $X$ and $Y$ are statistically independent), the information in the pair should be zero bit (since they are independent), but the entropy $H(X, Y)$ is maximal, and the distribution of partial entropy atoms reflects that (for a more detailed discussion of the PED in the context of maximum entropy systems, see [22] Supplementary Material).

Here, I will introduce a generalized decomposition of multivariate information that satisfies the intuitive understanding of what information is, does not require defining sources and targets, and which recovers the original, directed, PID as a special case. This generalized information decomposition (GID) is based on the decomposition of the Kullback-Leibler divergence [23], and the local partial entropy decomposition. This generalized information decomposition can be understood in a Bayesian sense as decomposing the information gained when one

updates their prior beliefs to a new posterior, and as a consequence, induces a decomposition of any information-theoretic metric that can be written as a Kullback-Leibler divergence (mutual information, total correlation, negentropy, etc). Being more general than the PID, it can also be used to decompose the information divergence between arbitrary distributions, as it does not enforce any particular constraints on the prior and the posterior.

First, I will introduce the necessary building-blocks (the Kullback-Leibler divergence, the local entropy decomposition), and then explore a special case to demonstrate the GID: the decomposition of the total correlation. Finally, I will discuss how the original PID of Williams and Beer can be re-derived and the possibility of future applications of this work.

## 1.1 A note on notation

This paper will make reference to multiple different kinds of random variables, at multiple scales, as well as multiple distributions. I will briefly outline the notational conventions used here. Probability distributions will be represented using blackboard font, typically using $\mathbb{Q}$ for a prior belief (in the context of a Bayesian prior-to-posterior update), and $\mathbb{P}$ for the posterior or a general probability distribution. We will use $\mathbb{E}_{\mathbb{P}(x)}[f(X)]$ to indicate the expected value operator of some function $f(x)$, computed with respect to the probability distribution $\mathbb{P}(X)$. Univariate random variables will be denoted with uppercase italics (e.g. $X$), multivariate random variables will be denoted with uppercase boldface (e.g. $\mathbf{X} = \{X_1, \ldots, X_N\}$). Specific (local) realizations of univariate or multivariate random variables will be denoted with their respective lowercase fonts (e.g. $X = x$ or $\mathbf{X} = \mathbf{x}$). Functions (e.g. the mutual information, the entropy, the Kullback-Leibler divergence, etc) will follow the same convention for expected and local values.

## 2 Background

Information, in the most general sense, refers to the reduction in uncertainty associated with observation. For example, consider rolling a fair, six-sided die. Initially, the value is unknown and all six values are equiprobable. However, upon learning that the value is *even*, three possibilities are immediately ruled out (the odd numbers one, three, and five), and the uncertainty about the value is decreased. The difference between the initial uncertainty and the final uncertainty after ruling out possibilities is the information about the state of the die that is disclosed by learning the parity of the state. Uncertainty about the state of a (potentially multidimensional) random variable is typically quantified using the Shannon entropy:

$$H(X) = -\sum_{x \in \mathfrak{X}} \mathbb{P}(x) \log \mathbb{P}(x) \tag{1}$$

Where $\mathbb{P}(x)$ is the probability of observing $X = x$. Upon gaining information (or reducing uncertainty), one is implicitly comparing two different probability distributions: a *prior* distribution (such as the initial uncertainty about the state of the dice) and a *posterior* distribution (the uncertainty about the state of the die after excluding the odd numbers). Following van Enk [24], one could heuristically describe information gained about $X$ generally as:

$$\text{Information}(X) = H^{\text{prior}}(X) - H^{\text{posterior}}(X) \tag{2}$$

The well-known Shannon mutual information is just a special case of this broader definition. The mutual information between $X$ and $Y$ can be written as:

$$I(X; Y) = H(X) - H(X|Y) \tag{3}$$

It is clear that that $H(X)$ is the $H^{\text{prior}}$, describing the initial beliefs about $X$ (i.e. that it is independent of $Y$). The second term $H(X|Y)$ is the $H^{\text{posterior}}$, describing the updated beliefs about $X$ after learning $Y$. The difference between these is the information gained when updating from a prior belief that $X \perp Y$ to the posterior based on the true joint distribution. The mutual information is a special kind of dependence between $X$ and $Y$, however; where the prior and posterior are related by the particular operation of marginalizing the joint. If one wanted a more general measure of information-gain for arbitrary priors and posteriors, they would need a different measure: the Kullback-Leibler divergence.

## 2.1 Kullback-Leibler divergence

For some multidimensional random variable $\mathbf{X} = \{X_1 \ldots X_N\}$, one can compute the information gained when updating from the prior $\mathbb{Q}(\mathbf{X})$ to the posterior $\mathbb{P}(\mathbf{X})$ with the Kullback-Leibler divergence:

$$D(\mathbb{P}||\mathbb{Q}) := \sum_{\mathbf{x} \in \mathfrak{X}} \mathbb{P}(\mathbf{x}) \log \frac{\mathbb{P}(\mathbf{x})}{\mathbb{Q}(\mathbf{x})}. \tag{4}$$

The $D(\mathbb{P}||\mathbb{Q})$ can be understood as the expected value of the log-ratio $\mathbb{P}(\mathbf{x})/\mathbb{Q}(\mathbf{x})$ (computed with respect to the posterior probability distribution $\mathbb{P}(\mathbf{X})$):

$$D(\mathbb{P}||\mathbb{Q}) = \mathbb{E}_{\mathbb{P}(\mathbf{X})}\left[\log \frac{\mathbb{P}(\mathbf{x})}{\mathbb{Q}(\mathbf{x})}\right]. \tag{5}$$

This can be re-written in explicitly information-theoretic terms by converting the log ratio into local entropies. Recall that, for some outcome $\mathbf{x} \in \mathfrak{X}$, the local entropy (or surprise) associated with observing $\mathbf{X} = \mathbf{x}$ is given by:

$$h^{\mathbb{P}}(\mathbf{x}) = -\log \mathbb{P}(\mathbf{x}). \tag{6}$$

The superscript $h^{\mathbb{P}}(\mathbf{x})$ denotes that the local entropy is being computed with respect to the distribution $\mathbb{P}(\mathbf{X})$, rather than $\mathbb{Q}(\mathbf{X})$. From this, simple algebra shows that:

$$D(\mathbb{P}||\mathbb{Q}) = \mathbb{E}_{\mathbb{P}(\mathbf{x})}\left[h^{\mathbb{Q}}(\mathbf{x}) - h^{\mathbb{P}}(\mathbf{x})\right]. \tag{7}$$

It is worth considering this in some detail, as it can help build intuition about what the Kullback-Leibler divergence really tells us. The term $h^{\mathbb{Q}}(\mathbf{x}) - h^{\mathbb{P}}(\mathbf{x})$ quantifies how much *more* surprised one would be to see $\mathbf{X} = \mathbf{x}$ if they were modelling $\mathbf{X}$ with the distribution $\mathbb{Q}(\mathbf{X})$ rather than $\mathbb{P}(\mathbf{X})$. This is obviously analogous to the intuitive definition given above in Eq 2, although this approach compares each of the local realizations of $\mathbf{X}$ first and then averaging, rather than averaging first and then subtracting. By Jensen's Inequality, this value must always be positive.

So far, I have discussed the multivariate random variable $\mathbf{X}$ as a single unit: the information about the *whole* is gained as a lump sum and contains very little insight into how that information is distributed over the various $X_i \in \mathbf{X}$. This is a significant limitation, as complex systems typically show a wealth of different information-sharing modes. For example, a natural question to ask might be; "what information gained is specific to $X_1$?" Or "what information gained is represented in the joint state of $X_1$ and $X_2$ together and no simpler combination of elements?" The standard machinery of classical information theory struggles to address these questions, and doing so rigorously requires leveraging recent developments in modern, multivariate information theory.

## 2.2 Partial entropy decomposition

To understand how information is distributed over the various components of **X**, I begin by describing the *partial entropy decomposition* (PED). The PED was first proposed by Ince [19], as an extension of the more well-known partial information decomposition (PID) that relaxes the requirement of an input/target distinction [6]. The PED begins with the same axiomatic foundation as the PID, but applies it to the multivariate entropy of a distribution, rather than the multivariate mutual information. Following its introduction, the PED was extensively explored by Finn and Lizier [20] (albeit under a different name), and more recently by Varley et al. in the context of inferring higher-order structure in complex systems [22].

For more details about the PED, see the cited literature, although I will provide a minimal introduction here. Consider a multivariate random variable $\mathbf{X} = \{X_1, \ldots, X_k\}$. The joint entropy $H(\mathbf{X})$ quantifies the average amount of information required to specify the unique state of **X**:

$$H(\mathbf{X}) = -\sum_{\mathbf{x} \in \mathfrak{X}} \mathbb{P}(\mathbf{x}) \log \mathbb{P}(\mathbf{x}) \tag{8}$$

This value is an expected value over the support set $\mathfrak{X}$: $H(\mathbf{X}) = \mathbb{E}_{\mathbb{P}(\mathbf{x})}[-\log \mathbb{P}(\mathbf{x})]$. For any individual realization **x** one can compute the *local entropy* (or surprisal) as $h(\mathbf{x}) = -\log \mathbb{P}(\mathbf{x})$. This value $h(\mathbf{x})$ quantifies how much uncertainty about **X** is resolved upon learning that $\mathbf{X} = \mathbf{x}$. From here on, I will describe the local partial entropy decomposition, although the logic is the same for the expected value as well, and local partial entropy atoms can be related to expected partial entropy atoms in the usual way.

The local entropy $h(\mathbf{X})$ is a scalar measure, describing the information content in **x** as a single entity and provides little insight into how that information is distributed over the structure of **x**. To get a finer-grained picture of how the various components of **x** contribute to $h(\mathbf{x})$, it would be useful to be able to elucidate how all the components of **x** share entropy. Formalizing this notion of "shared entropy" turns out to be non-trivial, however. Since the original introduction of the PID, various teams have proposed a plethora of redundancy functions that satisfy the Williams and Beer axioms and consequently induce the redundancy lattice. These different redundancy functions can return very different results making the problem of picking the "right" function a tricky one. For a partial review, see [25]. For didactic purposes it is sufficient to say that two (potentially overlapping) subsets $\mathbf{a}_1 \subset \mathbf{x}$, $\mathbf{a}_2 \subset \mathbf{x}$ share entropy if there is uncertainty about the state of the whole that would be resolved by observing either $\mathbf{a}_1$ alone or $\mathbf{a}_2$ alone.

For example, consider a playing card randomly drawn from a shuffled deck of 52 cards. If the player learns that the card is either a red card (belonging to the suits hearts of diamonds) or a face card (being a jack, queen, or king), the redundant entropy is the uncertainty about the card's identity that is resolved regardless of which of those two statements is true. In this case, the player can rule out the possibility that they are holding any card that is not red and not a face card (e.g. the two of clubs has been ruled out as a possibility). So, even though the player does not know which statement is true (red card or face card), and even though card colour and face are independent qualities, they have still gained information about their card.

Formally, one can define a *redundant entropy* function $h_\cap()$ that takes in some collection of subsets of **x** (often referred to as "sources") and returns the entropy redundantly shared by all of them. The seminal insight of Williams and Beer was that the set of collections of sources required to decompose **x** is constrained to the set of all combination of sources such that no

source is a subset of any other [6]:

$$\mathfrak{A} = \{\boldsymbol{\alpha} \in \mathcal{P}_1 \mathcal{P}_1(\mathbf{x}) : \forall \mathbf{a}_i, \mathbf{a}_j \in \boldsymbol{\alpha}, \mathbf{a}_i \not\subseteq \mathbf{a}_j\} \tag{9}$$

Where $\mathcal{P}_1$ is the power set function excluding the empty set $\emptyset$. This set of "atoms" is structured under the partial ordering relation:

$$\forall \boldsymbol{\alpha}, \boldsymbol{\beta} \in \mathfrak{A}, \boldsymbol{\alpha} \preceq \boldsymbol{\beta} \Leftrightarrow \forall \mathbf{b} \in \boldsymbol{\beta} \exists \mathbf{a} \in \boldsymbol{\alpha} \text{ s.t. } \mathbf{a} \subseteq \mathbf{b}. \tag{10}$$

This partial ordering is typically referred to as the *redundancy lattice* (see Fig 1). Given this structure, it is possible to uniquely specify the value of all $\boldsymbol{\alpha} \in \mathfrak{A}$ via Mobius inversion:

$$h_\partial^{\mathbf{x}}(\boldsymbol{\alpha}) = h_\cap^{\mathbf{x}}(\boldsymbol{\alpha}) - \sum_{\boldsymbol{\alpha}' \prec \boldsymbol{\alpha}} h_\partial^{\mathbf{x}}(\boldsymbol{\alpha}') \tag{11}$$

Finally, the sum of all the local partial entropy atoms reconstitutes the local entropy:

$$h(\mathbf{x}) = \sum_{\boldsymbol{\alpha} \in \mathfrak{A}} h_\partial^{\mathbf{x}}(\boldsymbol{\alpha}). \tag{12}$$

Just as the entropy is an expected value over local realizations, it is possible to compute the expected value of each atom over all configurations of $\mathbf{x} \in \mathfrak{X}$:

$$H_\partial^{\mathbf{X}}(\boldsymbol{\alpha}) = \mathbb{E}_{\mathbb{P}(\mathbf{X})}[h_\partial^{\mathbf{x}}(\boldsymbol{\alpha})] \tag{13}$$

To build intuition, consider a simple, two element system: $\mathbf{X} = \{X_1, X_2\}$, which draws states from $\mathbb{P}(\mathbf{X})$. The information contained in the realization $\mathbf{X} = \mathbf{x}$ can be decomposed into:

$$h(\mathbf{x}) = h_\partial^{\mathbf{x}}(\{x_1\}\{x_2\}) + h_\partial^{\mathbf{x}}(\{x_1\}) + h_\partial^{\mathbf{x}}(\{x_2\}) + h_\partial^{\mathbf{x}}(\{x_1, x_2\}) \tag{14}$$

and the marginal entropies can be similarly decomposed:

$$h(\mathbf{x}) = h_\partial^{\mathbf{x}}(\{x_1\}\{x_2\}) + h_\partial^{\mathbf{x}}(\{x_1\}) \tag{15}$$

$$h(\mathbf{x}) = h_\partial^{\mathbf{x}}(\{x_1\}\{x_2\}) + h_\partial^{\mathbf{x}}(\{x_2\}). \tag{16}$$

It is easy to see that the set $\{\{\{x_1\}\{x_2\}\}, \{\{x_1\}\}, \{\{x_2\}\}, \{\{x_1, x_2\}\}\}$ satisfies the requirements of Eq 9 and the ordering given by Eq 10.

Another way that the PED can be understood is as a special case of the partial information decomposition (PID). As discussed above, typically, the PID decomposes the mutual information that some set of variables disclose about a shared target: $I(X_1, \ldots, X_k; T)$. However, the PID can induce a PED in the particular case where the target is the joint state of all the inputs. If $\mathbf{X} = \{X_1, \ldots, X_k\}$, then $I(X_1, \ldots, X_k; \mathbf{X}) = H(\mathbf{X})$, and a PID of the mutual information will decompose the joint entropy of the whole $\mathbf{X}$ (this formulation of the PED was first noted by Makkeh et al., [26] and then later explored in detail by Varley et al., [22]). Intuitively, the PED can be understood as decomposing the information that the *parts* disclose about the *whole*.

As was previously mentioned, there have been a number of proposals for a natural functional form for $h_\cap$. The details of this debate are beyond the scope of this paper, although see [19, 20, 22] for three different approaches that satisfy the axioms required to induce the redundancy lattice. Given the relationship between the PID and PED described above, in theory, any redundant information function could be used to induce a PED, however, the GID imposes additional constraints. The most significant is that the redundancy function

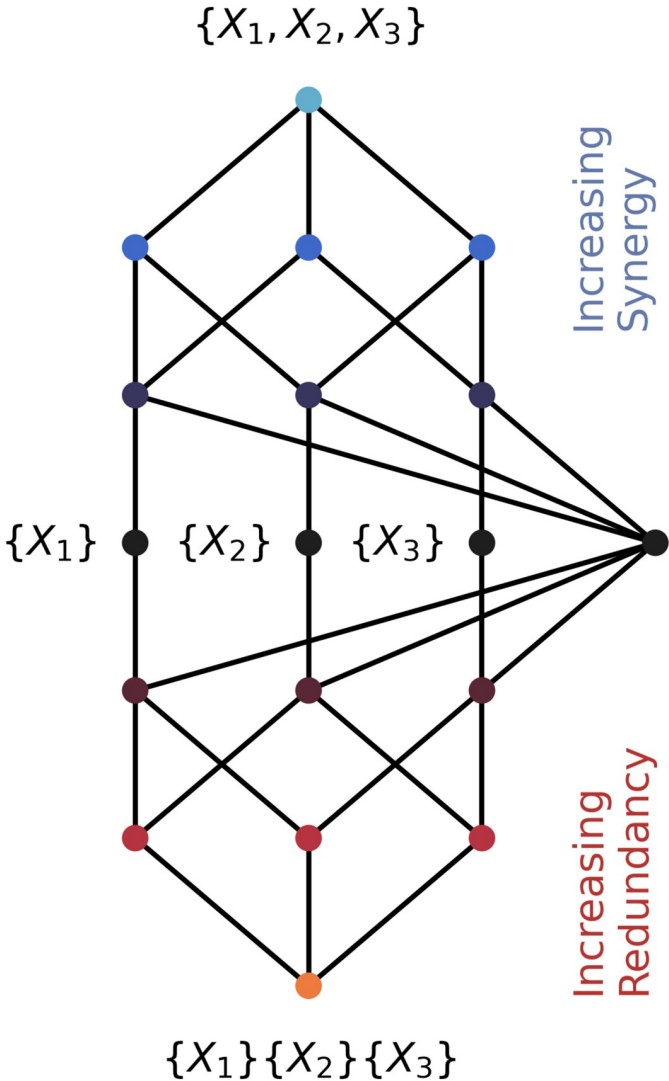

**Fig 1. The partial information lattice for three, interacting elements.** For three elements $X_1$, $X_2$, and $X_3$, the set of all sources is organized into a partially-ordered lattice structure. At the bottom of the lattice is the triple redundancy: $\{X_1\}\{X_2\}\{X_3\}$, which is the information that can be learned by observing $X_1$ alone or $X_2$ alone or $X_3$ alone. At the top is the triple synergy: $\{X_1, X_2, X_3\}$, which is the information can can only be learned by observing $X_1$, $X_2$, and $X_3$ together. The lattice represents a transition from redundancy-dominated interactions at the bottom (highlighted in red) towards synergy-dominated interactions at the top (highlighted in blue). In the middle are the three unique information atoms, corresponding to $X_1$ alone, $X_2$ alone, and $X_3$ alone.

must be localizable (i.e. the $I_{\min}$ function proposed by Williams and Beer will not work, as it is not fully localizable). All three existing redundant entropy functions ($h_{cs}$ [19], $h_{\min}$ [20] and $h_{sx}$ [22]) satisfy this property. Additionally, it would be helpful to require that the local partial entropy atoms be strictly non-negative. This rules out one of the three functions: $h_{cs}$ [19], being based on the local co-information, can return negative partial entropy atoms, which significantly complicates the interpretation of the PED and GID (discussed below). The two remaining functions, $h_{\min}$ [20] and $h_{sx}$ [22] do return strictly non-negative partial entropy atoms.

For didactic purposes, I choose the simplest of the three: the $h_{min}$ function proposed by Finn and Lizier [20]. For a collections of potentially overlapping sources $\boldsymbol{\alpha} = \{\mathbf{a}_1, \ldots, \mathbf{a}_k\}$:

$$h_{min}(\boldsymbol{\alpha}) = \min_i h(\mathbf{a}_i) \tag{17}$$

As mentioned above, the $h_{\min}$ function is just one possible redundant entropy function that satisfies the required axioms, and has its own costs and benefits. Different contexts may require different redundancy functions (for instance, $h_{\min}$ is not differentiable, while the closely related $h_{sx}$ is [26]).

## 3 Generalized information decompositions

We now have all the mathematical machinery required to introduce the generalized information decomposition. Recall from Eq 7 that the Kullback-Leibler divergence $D(\mathbb{P}||\mathbb{Q})$ can be written in terms of the expected difference in local entropies computed with respect to distributions $\mathbb{P}(\mathbf{X})$ and $\mathbb{Q}(\mathbf{X})$:

$$D(\mathbb{P}||\mathbb{Q}) = \mathbb{E}_{\mathbb{P}(\mathbf{x})}[h^{\mathbb{Q}}(\mathbf{x}) - h^{\mathbb{P}}(\mathbf{x})]$$

Given a localizable partial entropy decomposition (such as the one produced by $h_{min}$), it is possible to decompose each of the local entropies into its component atoms. For each atom, then, it is possible to compute the "partial Kullback-Leibler divergence"; the difference between the partial entropy atoms for each local realization computed with respect to the prior and the posterior:

$$D_\partial^{\mathbb{P}||\mathbb{Q}}(\boldsymbol{\alpha}) = \mathbb{E}_{\mathbb{P}(\mathbf{x})}[h_\partial^{\mathbb{Q}}(\boldsymbol{\alpha}) - h_\partial^{\mathbb{P}}(\boldsymbol{\alpha})]. \tag{18}$$

Note that I have extended the notation here: I must now indicate not only the Kullback-Leibler divergence from $\mathbb{Q}$ to $\mathbb{P}$, I must also indicate the specific atomic component of that information I am considering. In general, when referring to the atomic components of an information measure, I will use the $\partial$ subscript, and indicate which distribution a measure is computed with respect to which the relevant superscript.

For a three-element system $\mathbf{X} = \{X_1, X_2, X_3\}$, consider the "bottom" of the lattice: the triple-redundancy atom $D_\partial^{\mathbb{P}||\mathbb{Q}}(\{X_1\}\{X_2\}\{X_3\})$. It is the expected value of the difference between the local redundant entropies:

$$D_\partial^{\mathbb{P}||\mathbb{Q}}(\{x_1\}\{x_2\}\{x_3\}) = \mathbb{E}_{\mathbb{P}(\mathbf{x})}[h_\partial^{\mathbb{Q}}(\{x_1\}\{x_2\}\{x_3\}) - h_\partial^{\mathbb{P}}(\{x_1\}\{x_2\}\{x_3\})]. \tag{19}$$

Each partial entropy term quantifies how surprised one would be, regardless of whether they learned $X_1 = x_1$ or $X_2 = x_2$ or $X_3 = x_3$, computed with respect to probability distributions $\mathbb{Q}(\mathbf{x})$ and $\mathbb{P}(\mathbf{x})$ respectively. This difference represents how a change in beliefs (from prior to posterior) changes how surprised one would be to see a given configuration of elements: the information gain redundantly shared by all three variables.

There is no guarantee that the individual atomic components of the Kullback-Leibler divergence will be positive. Initially, the desire for a non-negative decomposition of multivariate information was such a core feature that non-negativity was included as a foundational requirement by Williams and Beer. Since then, however, the field has largely grown more comfortable with negative partial information atoms, and a number of proposed redundancy functions produce them (e.g. $I_{ccs}$ [27], $I_\pm$ [28], and $I_{sx}$ [26] for recent examples). When considering the generalized information decomposition in the case of the Kullback-Leibler divergence, the negativity is easily interpretable and not particularly strange. If, for example, $D_\partial^{\mathbb{P}||\mathbb{Q}}(\boldsymbol{\alpha}) < 0$,

then, on average, an observer would be more surprised to observe $\boldsymbol{\alpha}$ if they believe $\mathbb{P}$ (the posterior) rather than if they believe $\mathbb{Q}$ (the prior). In some sense, they have "lost" that specific information when they updated their beliefs. Since Jensen's inequality doesn't apply to the various $\boldsymbol{\alpha} \in \mathfrak{A}$, there's no *a priori* reason to assume non-negativity (although all atoms must sum to a non-negative number). This interpretation hinges on the non-negativity of local partial entropy atoms, however. If the local entropy atoms can be negative (as in the case of $h_{cs}$ [19]), the interpretation in terms of relative levels of surprisal is unclear. Consequently, I recommend that $h_{\min}$ or $h_{sx}$ be used in the context of the GID.

The GID inherits some of the limitations of the Kullback-Leibler divergence. One of the most salient is that it is only well-defined if the support set of $\mathbb{P}(\mathbf{X})$ (which I will denote as $\mathfrak{X}^{\mathbb{P}}$) is a subset or equal to the support set of $\mathbb{Q}(\mathbf{X})$ ($\mathfrak{X}^{\mathbb{Q}}$). If there are any $\mathbf{x}$ such that $\mathbb{P}(\mathbf{x}) > 0$ and $\mathbb{Q}(\mathbf{x}) = 0$, then the ratio $\mathbb{P}(\mathbf{x})/\mathbb{Q}(\mathbf{x})$ diverges. For didactic purposes, I generally assume that $\mathfrak{X}^{\mathbb{P}} = \mathfrak{X}^{\mathbb{Q}}$, although in the case of empirical data, this may not necessarily be true. In that case, there are a few options. The simplest is to simply not use the GID: in the case where $\mathfrak{X}^{\mathbb{P}} \not\subseteq \mathfrak{X}^{\mathbb{Q}}$, then there is a sense in which the relationship between the prior and posterior is fundamentally undefined. Alternately, one could add a small amount of noise to the data, so all the probability of all $\mathbf{x} \in \mathfrak{X}^{\mathbb{Q}} > 0$. This requires assuming that $\mathbf{x}$ *can* happen under distribution $\mathbb{Q}$, but that it is so unlikely that it was not observed in a finite-sized sample. Whether this assumption is valid or not depends on the particulars of the dataset in question, and will introduce significant bias to the final computation as $h^{\mathbb{Q}}(\mathbf{x})$ will be vastly larger than $h^{\mathbb{P}}(\mathbf{x})$ and consequently, the differences will dominate the expected value. Finally, one could define a restricted support set $\mathfrak{X}^*$ such that, for all $\mathbf{x} \in \mathfrak{X}^*$, $\mathbb{Q}(\mathbf{x}) > 0$. Those $\mathbf{x} \in \mathfrak{X}^{\mathbb{P}}$ but not in $\mathfrak{X}^*$ would be excluded and the probability distribution re-normalized. This will also introduce bias, as information in $\mathbb{P}(\mathbf{x})$ will inevitably be lost when those states are thrown out. All of these strategies have drawbacks, and any prospective scientist who finds themselves in such a situation should carefully consider the trade-offs involved: to what extent are the insights that could be gained from the GID in such a case compromised by the consequences of manipulating the underlying distributions? Ultimately, the decision must be handled on a case-by-case basis.

A large number of standard information-theoretic measures can be written in terms of the Kullback-Leibler divergence. Consequently, the decomposition presented above provides a considerable number of additional information decompositions "for free", as special cases in addition to the general case of arbitrary priors and posteriors. Here I will discuss one, the decomposition of the total correlation, in detail although other possibilities include the entropy production [29], the negentropy [30], and the classic bivariate mutual information. We will also very briefly discuss the cross-entropy, as it is a very commonly used metric in machine learning and artificial intelligence research.

### 3.1 Cross entropy decomposition

The cross entropy is a commonly used loss function in machine learning approaches [31]. For two distributions on $\mathbf{X}$, $\mathbb{P}(\mathbf{X})$ and $\mathbb{Q}(\mathbf{X})$, the cross entropy is defined by:

$$H^{\mathbb{P}\|\mathbb{Q}}(\mathbf{X}) := \mathbb{E}_{\mathbb{P}(\mathbf{X})}[-\log \mathbb{Q}(\mathbf{x})] \qquad (20)$$

Following the same logic as above, a decomposition of the cross-entropy is reasonably straightforward. It amounts to a local partial entropy decomposition on $H^{\mathbb{Q}}(\mathbf{X})$, and then the partial entropy atoms are aggregated across the different states using the distribution $\mathbb{P}(\mathbf{X})$ rather than $\mathbb{Q}(\mathbf{X})$. When used as a loss function in machine-learning applications, the decomposition of the cross entropy might be considered a "partial loss decomposition":

illuminating how the loss is distributed redundantly or synergistically over the features of a dataset.

## 3.2 Total correlation decomposition

Many information-theoretic quantities implicitly have an built-in prior distribution of maximum entropy (subject to some constraints). In the context of Bayesian inference and updating, there is a common argument that the most "natural" family of priors is the distribution that has the highest entropy. E.T. Jaynes argued for the "Principle of Maximum Entropy" [32], which posits that scientists should strive to use the least informative priors possible. This is a kind of formalization of Occam's Razor, suggesting that models of complex systems should not propose any more constraints on the space of possible configurations than is necessitated by the data in question. Intuitively, one can understand measures of deviation from independence as quantifying something like "how much more structured is this system than a kind of ideal gas." Here I will explore one of these multivariate information measures in the context of the generalized information decomposition: the total correlation.

Originally proposed by Watanabe [33] and later re-derived as the "integration" by Tononi and Sporns [34], the total correlation is one of three possible generalizations of the bivariate mutual information to arbitrary numbers of variables:

$$TC(\mathbf{X}) \coloneqq D\Big(\mathbb{P}(\mathbf{X})||\prod_{i=1}^{|\mathbf{X}|} \mathbb{P}(X_i)\Big). \tag{21}$$

Intuitively, $TC(\mathbf{X})$ can be understood as a measure of how much information is gained when modelling $\mathbf{X}$ based on its own joint statistics compared to if it is modelled as a set of independent processes (astute readers will remember this as equivalent to the intuition behind bivariate mutual information described above). One natural way to think about it is how many fewer yes/no questions an observer has to ask to specify the state of $\mathbf{X}$ based on the statistics of the "whole" compared to if each $X_i$ was resolved independently. It can be seen as a straightforward generalization of the more well-known definition of bivariate mutual information $I(X_1; X_2) = D(\mathbb{P}(X_1, X_2)||\mathbb{P}(X_1) \times \mathbb{P}(X_2))$. If ones considers the Bayesian interpretation of the Kullback-Leibler divergence, they can see that the prior in this case is the maximum-entropy distribution that preserves the marginal probabilities, and the posterior is the true distribution of the data.

For a given, potentially overlapping, set of sources $\boldsymbol{\alpha} = \{\mathbf{a}_1 \dots \mathbf{a}_k\}$, the partial total correlation $TC_\partial^{\mathbf{X}}(\boldsymbol{\alpha})$ quantifies how much of the total information gain is attributable to the particular collection of sources $\boldsymbol{\alpha}$, and crucially, no simpler combination of elements.

For a worked example, consider a three element system joined by a logical exclusive-or operator: $\mathbf{S} = \{X_1, X_2, T\}$, where $T = X_1 \oplus X_2$. We will begin by doing the PED of the prior distribution: the product of the marginals (which in this case is equivalent to the maximum-entropy distribution). The $h_{\min}$ redundancy function finds that the three bits of entropy are distributed equally over three atoms: one bit of information in the atom $H_\partial^{\mathbb{Q}}(\{X_1\}\{X_2\}\{T\})$, one bit of information in the atom $H_\partial^{\mathbb{Q}}(\{X_1, X_2\}\{X_1, T\}\{X_2, T\})$, and one bit information in the global synergy atom: $H_\partial^{\mathbb{Q}}(\{X_1, X_2, T\})$. See $H_\partial^{\mathbb{Q}}$ in Table 1. The next step is to do the PED on the true distribution given by the logical XOR gate. In this case, the PED finds two bits of entropy: one bit of redundancy $H_\partial^{\mathbb{Q}}(\{X_1\}\{X_2\}\{T\})$ and one bit in the atom $H_\partial^{\mathbb{Q}}(\{X_1, X_2\}\{X_1, T\}\{X_2, T\})$. Note that there is no information in the triple synergy atom in this case (see Table 1). When subtracting the two decompositions according to Eq 18, there remains 1 bit of information in the triple-synergy atom (see Table 1).

**Table 1. The partial entropy decompositions and partial total correlation decomposition.** Consider two distributions $\mathbb{P}(X_1, X_2, T)$ and $\mathbb{Q}(X_1, X_2, T)$. The distribution $\mathbb{Q}(\mathbf{S})$ is the maximum entropy distribution on three binary variables, while $\mathbb{P}(\mathbf{S})$ is the distribution of the logical-XOR gate (assuming equiprobable inputs).

| Atom | $H_\partial^\mathbb{Q}$ | $H_\partial^\mathbb{P}$ | $TC_\partial^\mathbf{S}$ |
|---|---|---|---|
| $\{X_1\}\{X_2\}\{T\}$ | 1.0 | 1.0 | 0.0 |
| $\{X_1\}\{X_2\}$ | 0.0 | 0.0 | 0.0 |
| $\{X_1\}\{T\}$ | 0.0 | 0.0 | 0.0 |
| $\{X_2\}\{T\}$ | 0.0 | 0.0 | 0.0 |
| $\{X_1\}\{X_2, T\}$ | 0.0 | 0.0 | 0.0 |
| $\{X_2\}\{X_1, T\}$ | 0.0 | 0.0 | 0.0 |
| $\{T\}\{X_1, X_2\}$ | 0.0 | 0.0 | 0.0 |
| $\{X_1\}$ | 0.0 | 0.0 | 0.0 |
| $\{X_2\}$ | 0.0 | 0.0 | 0.0 |
| $\{T\}$ | 0.0 | 0.0 | 0.0 |
| $\{X_1, X_2\}\{X_1, T\}\{X_2, T\}$ | 1.0 | 1.0 | 0.0 |
| $\{X_1, X_2\}\{X_1, T\}$ | 0.0 | 0.0 | 0.0 |
| $\{X_1, X_2\}\{X_2, T\}$ | 0.0 | 0.0 | 0.0 |
| $\{X_1, T\}\{X_2, T\}$ | 0.0 | 0.0 | 0.0 |
| $\{X_1, X_2\}$ | 0.0 | 0.0 | 0.0 |
| $\{X_1, T\}$ | 0.0 | 0.0 | 0.0 |
| $\{X_2, T\}$ | 0.0 | 0.0 | 0.0 |
| $\{X_1, X_2, T\}$ | 1.0 | 0.0 | 1.0 |

How does one interpret this? In the PED of the maximum entropy distribution, there is one bit of "synergistic entropy", since learning the state of any two variables isn't enough to fully resolve the state of **s**. If $X_1 = 0$, $X_2 = 0$, and $X_1 \perp X_2 \perp T$, then there are still two equiprobable states that **s** could be, so $h(\mathbf{s}|X_1 = 0, X_2 = 0)$ is maximal. It's only when all the parts are known can the whole be known (this is an intuitive example of "synergistic entropy" and its relationship to randomness). In contrast, for the logical XOR-gate, knowing any two variables is enough to specify the joint state of all three with total certainty. So, upon updating beliefs from the prior to the posterior, the single bit of synergistic entropy in the prior distribution is resolved. In the case of the XOR distribution, you do not need to know the state of all three elements to specify the whole, you only need the states of any two pairs of elements (which is reflected in the one bit of information in the atom $H_\partial^\mathbb{Q}(\{X_1, X_2\}\{X_1, T\}\{X_2, T\})$).

The decomposition of the total correlation into its atomic components can also be used to gain insight into the behaviour of measures that are derived from the total correlation. In fact, any measure that can be written in terms of total correlations can be decomposed into a linear combination of atomic components. Here I will discuss two, and in doing so, demonstrate how this decomposition can give us insights into the nature of higher-order information sharing.

**3.2.1 O-information.** The O-information is a heuristic measure of higher-order information-sharing in complex systems. It was first introduces as the "enigmatic information" by James et al. [35], and then later renamed the O-information and explored in much greater detail by Rosas, Mediano, and colleagues [30]. Given some multivariate random variable, the O-information of that variable, $\Omega(\mathbf{X})$, quantifies the extent to which the structure of $\mathbf{X}$ is dominated by redundant or synergistic information. If $\Omega(\mathbf{X}) > 0$, then the system is redundancy-dominated, while if $\Omega(\mathbf{X}) < 0$, the system is synergy-dominated. Since its introduction, the O-information has become an object of considerable interest: unlike the PID and PED, which

cannot be used for systems with more than four to five elements, the O-information scales much more gracefully, and has been applied to systems with hundreds of components [36–38].

The O-information was originally introduced as a difference between two different generalization of mutual information: the total correlation and the dual total correlation, however, recently Varley et al. [38], derived an equivalent definition in terms of solely total correlations:

$$\Omega(\mathbf{X}) \coloneqq (2 - N)TC(\mathbf{X}) + \sum_{i=1}^{N} TC(\mathbf{X}^{-i}) \tag{22}$$

By expanding each total correlation term into the associated linear combination of partial TC atoms and then simplifying, it is revealed that, for a three-variable system, the O-information can be understood as:

$$\begin{aligned}
\Omega(\{X_1, X_2, X_3\}) \quad &= 2 \times \{X_1\}\{X_2\}\{X_3\} \\
&+ 2 \times [\{X_1\}\{X_2\}) + \{X_1\}\{X_3\} + \{X_2\}\{X_3\}] \\
&+ 2 \times [\{X_1\}\{X_2, X_3\} + \{X_2\}\{X_1, X_3\} + \{X_3\}\{X_1, X_2\}] \\
&+ \{X_1\} + \{X_2\} + \{X_3\} \\
&+ 2 \times \{X_1, X_2\}\{X_1, X_3\}\{X_2, X_3\} \\
&+ \{X_1, X_2\}\{X_1, X_3\} + \{X_1, X_2\}\{X_2, X_3\} + \{X_1, X_3\}\{X_2, X_3\} \\
&- \{X_1, X_2, X_3\}
\end{aligned} \tag{23}$$

The notation has been simplified for visual clarity; each atom is a partial total correlation atom, representing the deviation from independence attributable to each set of sources.

There are several interesting things about this decomposition worth noting. The first is that terms of the form $TC_{\partial}^{123}(\{X_i, X_j\})$ do not appear. The O-information has previously been proved to be insensitive to bivariate dependencies [30], making it a "true" measure of higher-order dependency, and I propose that this is reflected in the absence of the bivariate partial total correlation atoms. The second thing to note is that this shows that O-information has a very strict definition of synergy and a comparatively relaxed definition of redundancy. The only atom that can ever count towards synergy is the very top of the lattice, as that is the information that is destroyed when any $X_i$ is removed from $\mathbf{X}$. Any information that is accessible from the remaining $\mathbf{X}^{-i}$ elements gets counted as "redundancy" (even if it involves three or more nodes). Consequently, one might argue that the O-information is *more* sensitive to redundancy than synergy, as there are simply more ways for information to be redundant than synergistic, particularly as $N$ grows. Future work on extensions of O-information that are more sensitive to lower-order synergies remains an open area of research.

**3.2.2 Tononi-Sporns-Edelman complexity.** The Tononi-Sporns-Edelman (TSE) complexity is one of the key developments in the study of applying multivariate information theory to complex systems. Initially proposed by Tononi, Sporns, and Edelman [34], for a given set of variables, the TSE complexity is hypothesized to quantify the balance between integration and segregation in the system. Formally, the TSE is highest when, on average, subsets of a system are statistically independent (i.e. the total correlation is zero), but the whole system itself strongly deviates from independence (i.e. the total correlation of the whole is high). This suggests a natural link to synergy: deviation from independence in the whole, but none of the parts at any scale. In the original presentation of the TSE complexity, Tononi, Sporns, and Edelman showed that the measure has a characteristic, inverted-U shape: when all elements are independent (global segregation), the complexity is low, and similarly, when all elements

are synchronized (global integration), the complexity is similarly low. Complexity is highest in an interstitial zone combining integration and segregation.

The TSE complexity can be written in terms of total correlations:

$$TSE(\mathbf{X}) = \sum_{i=1}^{N} \left[ \frac{i}{N} TC(\mathbf{X}) - \langle TC(\mathbf{X}^{\gamma_i}) \rangle \right] \quad (24)$$

Where the $\langle TC(\mathbf{X}^{\gamma_i}) \rangle$ refers to the average total correlation of every subset of $\mathbf{X}$ with $i$ elements.

Tononi, Sporns, and Edelman developed the TSE complexity almost a decade before Williams and Beer formalized the notions of redundancy and synergy; consequently the relationship between the two concepts has remained somewhat obscure. To the best of my knowledge, the first exploration of the relationship between TSE complexity and redundancy/synergy was by Rosas et al., in the initial introduction of the O-information [30], and then further explored by Varley et al., [38], who showed that the sign of the O-information was a function of the structure of the highest-level of the TSE bipartition hierarchy.

Since the TSE complexity can be written in terms of total correlations, it can be decomposed in the same manner as the O-information (see Eq 23). Once again, the partial-total correlation notation has been omitted for visual accessibility:

$$
\begin{aligned}
TSE(X_1, X_2, X_3) = \quad & -\{X_1\}\{X_2\}\{X_3\} \\
& -\frac{2}{3}\Big[\{X_1\}\{X_2\} + \{X_1\}\{X_3\} + \{X_2\}\{X_3\}\Big] \\
& -\frac{1}{3}\Big[\{X_1\}\{X_2, X_3\} + \{X_2\}\{X_1, X_3\} + \{X_3\}\{X_1, X_2\}\Big] \\
& +\frac{1}{3}\Big[\{X_1, X_2\}\{X_1, X_3\} + \{X_1, X_2\}\{X_2, X_3\} + \{X_1, X_3\}\{X_2, X_3\}\Big] \\
& +\frac{2}{3}\Big[\{X_1, X_2\} + \{X_1, X_3\} + \{X_2, X_3\}\Big] \\
& +\{X_1, X_2, X_3\}
\end{aligned}
\quad (25)
$$

Here, in the three-variable case, an elegant pattern is revealed by the decomposition: as one travels farther down the bottom half of the redundancy lattice (see Fig 1 for reference), the information in each atom becomes increasingly penalized (in 1/N increments), while as one travels farther up the upper half of the lattice, each atom becomes increasingly "rewarded." A moment's reflection shows that this broadly consistent with the original intuition put forward by Tononi et al.,: the presence of redundant information shared by many single variables indicates that the elements at the micro-scale are not segregated, and so that information counts against the TSE complexity. In contrast, synergy reflects a kind of global integration, and so it positively contributes to TSE.

While this may seem like a fairly banal rephrasing of the original intuition behind TSE, further consideration suggests that this tells us something interesting about synergy: if the TSE is low when integration or segregation dominate and high when both are in balance (see Tononi et al., Fig 1D [34]), then this suggests that synergy is not merely another "kind" of integration, but rather is itself a reflection of a system balancing both integration and segregation. Since increasingly higher-order synergy drives up TSE, it follows that increasingly higher-order deviations from independence must also imply a balance of integration and segregation. This is consistent with recent empirical findings; in analysis of human neuroimaging data, synergy has been repeatedly found to sit "between" highly-integrated "modules" in the brain, while

redundancy is higher within the modules [22, 38], suggesting that synergy forms a kind of "shadow structure": a network of higher-order dependencies that are largely invisible to the standard techniques of network science and functional connectivity.

### 3.3 Recovering single-target PID

As previously mentioned, the standard Shannon mutual information is a special case of the more general Kullback-Leibler divergence. Consequently, one would expect that the generalised information decomposition should recover the classic single-target PID. There are a number of ways to write out the bivariate mutual information in terms of a Kullback-Leibler divergence, but the most salient one for the purposes of this paper is the definition:

$$I(X_1, X_2; T) := \mathbb{E}_T[D(\mathbb{P}(X_1, X_2 | T = t) || \mathbb{P}(X_1, X_2))] \tag{26}$$

For each value of $t$ that $T$ can adopt, there is a Kullback-Leibler divergence between the prior $\mathbb{P}(X_1, X_2)$ to the posterior $\mathbb{P}(X_1, X_2 | T = t)$. Each of these divergences can be decomposed into four GID atoms (corresponding to redundancy, unique information, and synergy respectively), and then the expected value of each atom computed with respected to $\mathbb{P}(T)$. The decomposition induced is analogous to the informative/misinformative decomposition first explored by Finn and Lizier [28] and later expanded on by Makkeh et al., [26]. The GID in this case follows the expected form:

$$I(X_1, X_2; T) = D_\partial^{12T}(\{X_1\}\{X_2\}) + D_\partial^{12T}(\{X_1\}) + D_\partial^{12T}(\{X_2\}) + D_\partial^{12T}(\{X_1, X_2\}). \tag{27}$$

$$I(X_1; T) = D_\partial^{12T}(\{X_1\}\{X_2\}) + D_\partial^{12T}(\{X_1\}). \tag{28}$$

$$I(X_2; T) = D_\partial^{12T}(\{X_1\}\{X_2\}) + D_\partial^{12T}(\{X_2\}). \tag{29}$$

If one uses the $H_{sx}$ measure, the resulting decomposition is equivalent to the PID computed using $I_{sx}$, and likewise if one uses $H_{min}$, the resulting decomposition is equivalent to $I_\pm$ [28]. An intriguing feature of the GID is that different ways of formalizing the mutual information can actually induce different decompositions. Consider an alternative definition of the mutual information, also expressed in terms of a Kullback-Leibler divergence:

$$I(X_1, X_2; T) = D(\mathbb{P}(X_1, X_2, T) || \mathbb{P}(X_1, X_2) \times \mathbb{P}(T)). \tag{30}$$

This decomposition will result in eighteen atoms (since it describes a three-element system). The sum of all the atoms will still be the mutual information $I(X_1, X_2; T)$, but the way that information is assigned to different combinations of elements is entirely different. Furthermore, there is not, at present, an obvious way of linearly combining the eighteen atoms generated by Eq 30 to recover the expected four atoms seen above (although I cannot say that it is impossible either, merely that if it does exist, it escapes this author). How can one make sense of this unusual feature? One possible explanation is that, while both formulations (Eqs 26 and 30) are the same mutual information, they implicitly privilege different ways of thinking about the relationship between $X_1$, $X_2$, and $T$. In the first case (Eq 26), the system under study is naturally understood as the two-dimensional pair of $X_1$ and $X_2$: the target variable $T$ is "external" in some sense and modulates the behaviour of $X_1$ and $X_2$ from outside the system. In contrast, in the case of Eq 30, the natural perspective is of a three-element system $X_1$, $X_2$, and $T$, which are decomposed together, producing eighteen atoms. Currently, this is merely a hypothesis that attempts to explain this discrepancy, but it suggests that the perspectives or biases of the

observer/analyst deploying an information-theoretic tool can inform on the apparent structure of the system under study.

## 4 Discussion

In this paper, I have introduced a generalized information decomposition (GID), based on the Kullback-Leibler divergence and the local partial entropy decomposition (PED). This GID allows a decomposition of any information gain that occurs when updating from a distribution of prior beliefs to a new posterior distribution. As a consequence, a significant number of information-theoretic metrics can be studied using the GID, including the classic single-target multivariate mutual information, the total correlation, the negentropy, the cross entropy, and more. This decomposition is consistent with the fundamental intuitions about "what is information", and unlike the classic PID, does not require defining classes of "inputs" and "targets." In this final section, I will discuss the implications and possible applications of the GID to the analysis of complex systems.

### 4.1 Many different synergies

The most obvious take-away from this analysis is that a given distribution $\mathbb{P}(\mathbf{X})$ can have many different "kinds" of redundancy or synergy, depending on exactly what measure is being decomposed. We have discussed the partial entropy, the partial total correlation, and multiple kinds of partial mutual information. While some of these are interconvertible (for example, Ince showed that the PID can be written in terms of sums and differences of PED atoms [19]), others do not appear to be directly interconvertible (such as the two different decompositions of mutual information discussed earlier). For a given probability distribution, depending on how exactly one wishes to decompose it, entirely different distributions of redundancies and synergies can be extracted; some may even have different signs. This means that, going forward, when analysing higher-order information in complex systems, care must be taken to specify exactly how concepts like redundancy and synergy are being defined, and more importantly, how they should be interpreted. Conflating the partial entropy term $H_\partial^{12T}(\{X_1\}\{X_2\}\{X_3\})$ with the partial total correlation term $TC_\partial^{12T}(\{X_1\}\{X_2\} \times \{X_3\})$ or the partial divergence term $D_\partial^{\mathbb{P}\|\mathbb{Q}}(\{X_1\}\{X_2\}\{X_3\})$ may lead to confusion or misinterpretation.

There is precedent for such a landscape of possibilities: the PID has long struggled with the problem that multiple redundancy functions can satisfy the fundamental axioms, while inducing totally different decompositions of a given mutual information [21, 25]. While initially seen as a problem, some have argued for a perspective of "pragmatic pluralism" [22], and that the different options may have distinct and complementary use-cases for building a complete picture of a given system. A similar argument can be made here: depending on the specific system being analysed, different information decompositions may be more or less appropriate. If there is a well-defined notion inputs and targets, such as when studying directed information flows in neural systems [9], or how multiple social identities synergistically inform on a single outcome [39], then a single-target PID may be the most appropriate. In contrast, if one is looking at higher-order generalizations of undirected functional connectivity [22, 38], then a PED or GID could be more relevant. Different combinations of directed and undirected decompositions, coupled with different definitions of redundancy, creates an very rich field of possibilities that could be applied to a variety of different complex systems, at many scales.

## 4.2 Towards a unified theory of multivariate information decomposition

This proposal for a generalized decomposition of multivariate information is one of several different recent attempts to generalize the PID. As previously discussed, the first approach was the PED [19, 20], which relaxed the source/target distinction by decomposing the entropy instead of the mutual information.

An interesting consequence of the GID is that it reveals fundamental connections between (almost) all of the existing information decomposistion approaches: PID, PED, and GID (see Fig 2). The paper began with the single-target PID, and from that one can construct the undirected PED (by setting the target to be the joint state of all the elements). The GID is then constructed from local PEDs, and admits single-target PID as a particular special case, completing the cycle. Each of the approaches can be constructed from, or is a special case of, the others. This unifies the directed and undirected decompositions of both information and entropy into a coherent foundation, on which future developments may be constructed.

However, not all information decomposition approaches have been reconciled yet. Another approach to generalizing the PID was the integrated information decomposition (ΦID) from Mediano et al., [40]. In contrast the PED, the ΦID still requires dividing a system into "inputs" and "targets", but it relaxes the requirement of only having a single target. The ΦID can accept an arbitrary number of inputs and targets. This makes it particularly natural for analysing temporal dynamics: in such a case, the inputs are the states of all the elements at time $t$, and the targets are the states of the elements at time $t + \tau$. Application of the ΦID to clinical data has

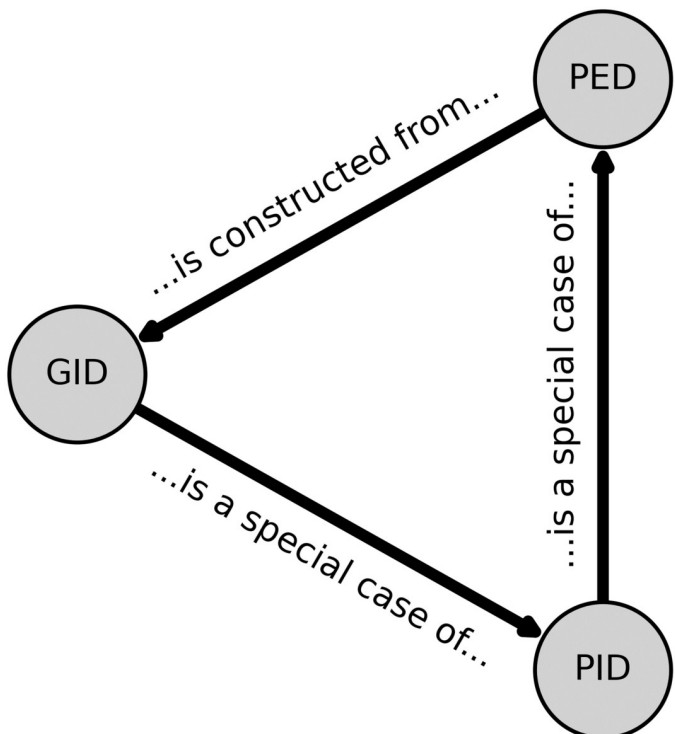

**Fig 2. The relationship between PID, PED, and GID.** Each of the three information decompositions (the PID, PED, and GID) can be related to each-other in terms of how one is constructed from another. The PID is a special case of the GID (when the prior is the product of the marginals of the posterior), the PED is a special case of the PID (when the target is the joint-state of all the inputs), and the GID is constructed from the (local) PED. None of the three are fundamentally "prior" to any other, and this relationship forms the beginning of a unified theory of multivariate information decomposition.

shown that the distributions of temporal redundancies and synergies tracks level of consciousness [16, 17], and analysis of spiking neural dynamics has found the distribution of redundancies and synergies varies over the course of neuronal avalanches [41].

As it currently exists, the ΦID does not fit into the GID schema described here, as it uses a different lattice structure as a scaffold and is not currently formalized in terms of a Kullback-Leibler divergence. I conjecture, however, that there should be a way to reconcile these approaches and further generalize the existing GID to account for the ΦID, although this problem is beyond the scope of the current paper.

Finally, the most recent approach to generalizing the PID was proposed by Gutknecht et al. [42]. This approach generalizes the notion of a "base concept" in information decomposition (such as redundancy) and reveals the general logical structure of the different possible single-target PIDs that can exist. Conceivably, any one of these base-concepts could be applied to the GID presented here, although the resulting interpretations will vary. Since the PED can always be defined as a PID of the "parts" onto the "whole", any PID based on a base-concept such as redundancy, weak synergy, or vulnerable information could conceivably induce a PED, and if that PED is localizable, a subsequent decomposition of the Kullback-Leibler divergence. This variety of base concepts can be added to the already rich set of possibilities (multiple redundancy functions, multiple decompositions) to expand the set of tools that scientists can use for attempting to analyse and model complex systems.

## 4.3 Applications of the GID

This paper has focused on the decomposition of the total correlation as a case study application of the more general decomposition. We chose to focus on the total correlation due to its links to the O-information and the Tononi-Sporns-Edelman complexity, as well as because it is a measure that most information theorists are familiar with. However, there are many other applications that would be worth exploring. For instance, one area of future work that may be of interest is the decomposition of the entropy production, which uses the Kullback-Leibler divergence to estimate the temporal irreversibility of a dynamic system [29]. Recently, Lynn et al., introduced a decomposition of the entropy production, although it is based on a different logic than the partial information decomposition and does not include a notion of redundancy [43]. Luppi et al., recently introduced their own decomposition of temporal irreversibility, although it is not based on the entropy production [44]. The GID allows for a decomposition of the entropy production within the well-known framework of the antichain lattice. Unlike Lynn et al.'s approach, it does not require making assumptions that no two variables change state at the same time [43] and allows for a distinction between higher-order redundancies and synergies.

Finally, this approach may be very useful to cognitive scientists interested in how agents equipped with multiple sensory channels navigate complex, multidimensional environments. Any agent attempting to survive in such a world must learn the statistical regularities of its environment; regularities that may be redundantly or synergistically distributed across different sensory modalities. The Kullback-Leibler divergence is a key feature of many Bayesian approaches to theoretical neuroscience and cognitive science (e.g. the Free Energy Principle [45]), and often used to describe the process by which an agent updates its internal world-model: in these approaches, the world model at time $t$ is the prior and it is updated in some Bayesian fashion to a posterior after some interaction with the external world has occurred. Having the ability to finely decompose information may give insights into how agents learn and exploit potentially higher-order correlations in their environments. For example, some sensory inputs may be redundant (taste and smell, for example, could both be equivalently

informative about whether a food is spoiled), while others may be unique (the sound of a tiger growling in the dark is highly informative on its own), and some may even be synergistic (Luppi et al., use stereoscopic depth perception as an example of an emergent synergy between two channels, in this case, the right and left eyes [17]). Recent theoretical work in group dynamics has suggested that maximizing synergy can have significant benefits for collectives attempting to survive in complex environments [46]. It remains an open question whether there is a similar incentive to maximize the representation of higher-order synergies in one's world model, although such a hypothesis seems reasonable. Taking an information-decomposition approach to the problem of free-energy minimization may provide rich insights into how agents navigate and thrive in our complex, interconnected world.

## 5 Conclusions

This paper has discussed a generalization of the single-target partial information decomposition that relaxes the requirement that elements be grouped into "inputs" and "targets", while still preserving the basic intuitions about information. Based on the Kullback-Leibler divergence and the local partial entropy decomposition, this generalized information decomposition can applied to any information gained when updating from a set of prior beliefs to a new posterior. This generality implies that any information-theoretic measure that can be written as a Kullback-Leibler divergence admits a decomposition, such as the total correlation, negentropy, mutual information, and more. The generalized information decomposition could be of great utility in understanding the mereological relationships between "parts" and "wholes" in complex systems.

## Acknowledgments

TFV would like to thank Ms. Maria Pope, Dr. Olaf Sporns, and Dr. Josh Bongard for their support in producing this research.

## Author Contributions

**Conceptualization:** Thomas F. Varley.

**Formal analysis:** Thomas F. Varley.

**Investigation:** Thomas F. Varley.

**Writing – original draft:** Thomas F. Varley.

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
