## [Decision Letter · Decision Letter 0]

2 Nov 2023

PONE-D-23-29949Generalized decomposition of multivariate information.PLOS ONE

Dear Dr. Varley,

Thank you for submitting your manuscript to PLOS ONE. After careful consideration, we feel that it has merit but does not fully meet PLOS ONE’s publication criteria as it currently stands. Therefore, we invite you to submit a revised version of the manuscript that addresses the points raised during the review process. We have received your inquiry about the manuscript status and the possibility of publishing the paper before the end of the year. We will keep this deadline in mind and process the manuscript as quicky as possible. Please refer to the attached reviewer comments for suggested improvements of the manuscript. In particular, the major questions raised by Reviewer 2 on synergy interpretation in TSE complexity and the GID of I(X1, X2 : Y) should be addressed in the revised version of the manuscript. Please also consider the comment by Reviewer 1 on the comment that the proposed measure is unrelated to the notion of statistical independence, which warrants some discussion.

We look forward to receiving your revised manuscript.

Kind regards,

Patricia Wollstadt, Ph.D.

Academic Editor

PLOS ONE

 [The funders had no role in study design, data collection and analysis, decision to publish, or preparation of the manuscript.]. 

5. "Please upload a copy of Figure 1D, to which you refer in your text on page 10. If the figure is no longer to be included as part of the submission please remove all reference to it within the text.

Reviewers' comments:

Reviewer's Responses to Questions

**Comments to the Author**

1. Is the manuscript technically sound, and do the data support the conclusions?

Reviewer #1: Yes

Reviewer #2: Yes

2. Has the statistical analysis been performed appropriately and rigorously? 

Reviewer #1: N/A

Reviewer #2: N/A

3. Have the authors made all data underlying the findings in their manuscript fully available?

Reviewer #1: Yes

Reviewer #2: Yes

4. Is the manuscript presented in an intelligible fashion and written in standard English?

Reviewer #1: Yes

Reviewer #2: Yes

5. Review Comments to the Author

Reviewer #1: This paper describes how a local partial entropy decomposition can be applied to decompose a KL-divergence, and so other measures calculated from KL-divergences.

The paper is very clearly written and explained.

Major comments

Really nice didactic introduction. Could maybe extend this a bit to expand on the PED and give some more examples of the different measures (not at all necessary, just may make the paper useful to a wider range of readers).

I have a practical concern about calculating local measures on different distributions: how to handle symbols with zero probability in one of P or Q and non-zero probability in the other (e.g. equation 18). It would be nice to see this discussed.

I appreciate the bulk of the paper is a theoretical contribution which is very clear, but I think when presenting hmin and the example based on that it would be good to be more upfront about the limitations and interpretation of this measure. I.e. by definition it is quantifying the minimum amount local surprisal, and therefore gives a quantity that is unrelated to the notion of statistical independence (which is maybe counter intuitive given the notion of redundancy and synergy as types of interaction). I think could at least referent the similar problems with Imin, the two-bit copy problem etc. For example, one might expect the entropy decomposition for three independent bits in Table 1 would be {1}={2}={T} = 1. This would match the notion of statistical independence and that each contributes independent and unique uncertainty to the overall joint distribution. It is much harder to understand why there is one bit of synergy in this system – for example if there were three completely unique coins from different currencies tossed in different capital cities: for example a €1 coin tossed in Paris, a £1 coin tossed in London a year later, and a quarter tossed in New York a year later, it is hard to see how to interpret a decomposition that says these three events share a synergistic interaction in their uncertainty. Following the explanation at the bottom of page 7, “It’s only when all the parts are known can the whole be known”, this means there can never be unique entropy – any variable with unique entropy of course means that the full entropy can only be known when that variable is added to the others, and so would be counted as synergy according to this interpretation. So in my view this example is a bit problematic – I would suggest either including several more examples, perhaps using different measures, to give more of a view of how it works with different system and possibly different measures, or leaving out this example (because the interesting theoretical links stand alone regardless of the specific measure).

Minor comments

Abstract: “in as a” -<> “as a”

P4, Para “The local entropy h(X)”, should be lower case X?

P5, after eq 11. “sub” -> “sum”

P7, last para, If X1 = 0 and X2 = 0 (and lines below). Might be getting confused with notation, but should these not be lower case?

It would be nice to see some citations to earlier practical applications of PID in neuroimaging e.g. 10.1371/journal.pbio.2006558, 10.1016/j.cub.2019.04.067 (this one actually uses PID in 3 different ways, to justify use of single sources, for comparison of encoding models, and for comparison of decoding from the predictions of encoding models).

Reviewer #2: The manuscript is technically sound but it has one mathemtical issue that doesn't affect its major results (Please find the details in the Reviewer Attachments). In principle the manuscript is well written however parts of it could benefit from restructuring. Please find the our full comments and assessment in the Reviewer Attachments.

6. PLOS authors have the option to publish the peer review history of their article (what does this mean?). If published, this will include your full peer review and any attached files.

Reviewer #1: No

Reviewer #2: No

---

## [Decision Letter · Decision Letter 1]

29 Dec 2023

Generalized decomposition of multivariate information.

PONE-D-23-29949R1

Dear Dr. Varley,

We’re pleased to inform you that your manuscript has been judged scientifically suitable for publication and will be formally accepted for publication once it meets all outstanding technical requirements.

Kind regards,

Patricia Wollstadt, Ph.D.

Academic Editor

PLOS ONE

Additional Editor Comments (optional):

Reviewers' comments:

Reviewer's Responses to Questions

**Comments to the Author**

1. If the authors have adequately addressed your comments raised in a previous round of review and you feel that this manuscript is now acceptable for publication, you may indicate that here to bypass the “Comments to the Author” section, enter your conflict of interest statement in the “Confidential to Editor” section, and submit your "Accept" recommendation.

Reviewer #1: (No Response)

Reviewer #2: All comments have been addressed

2. Is the manuscript technically sound, and do the data support the conclusions?

Reviewer #1: Yes

Reviewer #2: Yes

3. Has the statistical analysis been performed appropriately and rigorously? 

Reviewer #1: Yes

Reviewer #2: N/A

4. Have the authors made all data underlying the findings in their manuscript fully available?

Reviewer #1: Yes

Reviewer #2: Yes

5. Is the manuscript presented in an intelligible fashion and written in standard English?

Reviewer #1: Yes

Reviewer #2: Yes

6. Review Comments to the Author

Reviewer #1: The author has revised the manuscript to address all the comments, I only have minor comments. I leave it up to them whether to make any minor changes to address this.

On negative partial entropy atoms. Regardless of choice of measure (i.e. implied just by the PED lattice), the existence of mechanistic information reduncancy (i.e. redundant information from PID in a system where the inputs are marginally independent – e.g. the AND gate) necessarily implies the existence of a negative PED term ({1}{2} must be negative in the presence of mechanistic information redundancy). See Ince PED arXiv. On the discussion of non-negativity it would be useful to include this point I think. How are the non-negative entropy decompositions compatible with the corresponding PIDs for AND (which mostly show non-zero redundancy).

I appreciate the explanation regarding your interpretation of synergy, maybe something to discuss offline. I think synergy should be the information about the target that can *only* be learned by observing all three coins *together* (by definition, an interaction representing a deviation from statistical independence), and not determined by three individual observers each making their best prediction based on their single observed coin, and someone else combining the predictions. If it is the target can only be learned by all the inputs, then this is a much weaker statement that loses this important “emergent” property that the term “synergy” is usually used to refer to. This weaker version clearly applies to any system/events without redundancy. So in this framework there is no notion of independence anymore (this was the key point of my original comment which wasn’t addressed). In what situations can you have unique entropy? You have shared entropy (redundant) or not shared (independence, which is classes as synergy because you only know the full output if you know the values of the independent components). I think it would be more useful perhaps to give this a separate name, it is not synergy (ie whole more than sum of parts, by definition since it ignores the sum of parts), but instead maybe “Non-redundant”. If the two-bit copy is not the canonical example of unique entropy/information in this framework, it would be very useful to provide a simple system that is, if not purely “unique”, what two bit system maximizes the “unique” entropy in this framework, and what is that maximum acheiveable value (presumably less that 1bit). If this framework cannot measure unique entropy, maybe the decomposition could be simplified to just have two terms (redundant vs non-redundant) – this would then make more sense to be in terms of two-bit copy etc.

Reviewer #2: Please see the attached file for minor revisions. We only have a few suggestions for improvement. It is to be decided by the author wether to take them into consideration. Please let me know if the file is not there since I tried couple of time to uploaded but not sure if it was successful.

7. PLOS authors have the option to publish the peer review history of their article (what does this mean?). If published, this will include your full peer review and any attached files.

Reviewer #1: No

Reviewer #2: No

---

## [Editor Report · Acceptance letter]

25 Jan 2024

PONE-D-23-29949R1 

PLOS ONE

Dear Dr. Varley, 

I'm pleased to inform you that your manuscript has been deemed suitable for publication in PLOS ONE. Congratulations! Your manuscript is now being handed over to our production team.

Kind regards, 

on behalf of

Dr. Patricia Wollstadt 

Academic Editor

PLOS ONE